# Factors associated with the use of long-lasting insecticidal nets in pregnant women and mothers with children under five years of age in Gaza province, Mozambique

Amancio Vicente Nhangave[1,2], Isabelle Munyangaju[3]*, Dulce Osório[3], Edy Nacarapa[3], Sozinho Ndima[4], Alfa Moiane[5], Ismail Chiposse[1], Izaidino Muchanga[6], Abuchahama Saifodine[2,5]

1 Gaza Provincial Research Nucleus, Provincial Health Directorate, Xai-Xai, Mozambique, 2 Faculty of Medicine, Eduardo Mondlane University, Maputo, Mozambique, 3 Tinpswalo Association, Vincentian Association to Fight AIDS and TB, Gaza Province, Mozambique, 4 Faculty of Medicine, Department of Community Health, Eduardo Mondlane University, Maputo, Mozambique, 5 National Malaria Control Programme, Gaza Provincial Health Directorate, Xai-Xai, Mozambique, 6 Faculty of Health Science, University of Saint Thomas, Gaza Campus, Gaza, Mozambique

* imunyangaju@gmail.com

## Abstract

Malaria remains a major public health concern worldwide. Malaria is endemic in Mozambique, with seasonal fluctuations throughout the country. Although the number of malaria cases in Mozambique have dropped by 11% from 2020 to 2021, there are still hotspots in the country with persistent high incidence and low insecticide-treated bed net usage. The aim of this study is to evaluate the factors associated with the use of long-lasting insecticidal nets by pregnant women and women with children under 5 years old in two hotspot districts in the Gaza province, Mozambique. A descriptive, qualitative cross-sectional study was conducted between June 15th and 21st 2022. An in-depth interview process was conducted with pregnant women and mothers with children under five years old, exploring their beliefs, experiences, and perception of messages conveyed by health professionals when long-lasting insecticidal nets were being supplied. A total of 48 women participated (24 pregnant women and 24 women with children under 5 years). Most participants recognized the protective effects of long-lasting insecticidal nets in preventing malaria, and understood that women and children were high risk groups. The nets were reported to cause side effects and difficulty breathing by 100% of pregnant women, while 54.2% of mothers with children under 5 reported no side effects. The majority of women in both groups reported that their health professionals did not educate them about how to use or handle the nets properly. Only 16.7% of mothers with children under 5 received correct handling instructions. Providing clear, culturally sensitive, and practical information on the correct use of LLINs, as well as regular monitoring of their proper use, would be a great step forward for Mozambique's national malaria program.

**Data Availability Statement:** The datasets analysed during the current study are available and have been submitted as supplementary information, and within the manuscript.

**Funding:** The author(s) received no specific funding for this work.

**Competing interests:** The authors have declared that no competing interests exist.

## Introduction

Malaria remains a major public health concern worldwide. According to the World Health Organization (WHO), an estimated 247 million cases of malaria and 619,000 deaths occurred worldwide in 2021. There were 95% of cases and deaths in the WHO African region alone, and 55% of all cases and 50% of deaths occurred in 6 countries in the region, including Mozambique [1].

Malaria is endemic in Mozambique, with seasonal fluctuations throughout the country. There is a wide variation in malaria prevalence across provinces, with higher prevalence rates in northern provinces (e.g. 57.3% in Cabo Delgado province) than southern provinces (e.g. 1% in Maputo city, 16.9% in Gaza province). There was also a high prevalence of malaria in rural areas and among children in Mozambique. Children under 5 years of age and pregnant women are considered high-risk groups [2]. The number of malaria cases in Mozambique dropped by 11% from 11.3 million in 2020 to 10.6 million in 2021 following targeted campaigns [3].

In order to combat malaria, Mozambique distributes large amounts of long-lasting insecticidal nets (LLINs) to all districts every three years. However, the country still ranks fourth among 11 countries globally with the highest malaria burden in terms of estimated cases and deaths [1].

According to the national Malaria Control Strategic Plan 2017–2022, 85% of the population should be covered with at least one vector control intervention (indoor residual spraying and use of LLINs) in all districts [4]. According to the Malaria Indicators Survey (Inquérito Nacional sobre Indicadores de Malária, IIM) conducted in Mozambique in 2018, 18% of respondents reported not sleeping under a LLINthe night before the survey. The majority had no LLIN at home, while 15% said they didn't use the net because there weren't any mosquitoes and 10% didn't like it [5].

It was found in IIM 2018 that Gaza province had the second lowest LLIN usage percentage, around 66.8%, with at least one LLIN for two people at the time of the survey, while Cabo Delgado province had the highest rate of 88.8%. Among children under 5, Gaza had a lower percentage of children who slept under LLINs before the survey, with 66.3%, compared with Cabo Delgado province, with 92.9% [5].

In a study by Wetzler et al (2022), researchers showed that due to the mass distribution of LLINs an estimated 14,040 under 5 child deaths were averted between 2012 and 2018 due to increase coverage [6]. However, it is vital that this coverage is accompanied by education on the benefits, correct use and handling of LLINs in order to build on the gains reached up to date. More and more studies are reporting on the nefarious effects of lack of awareness on the proper use of LLINs. Populations have been reported in East Africa and northern Mozambique to be using LLINs for fishing thereby harming the environment and negatively impacting the economy for short term economic gains [7–9].

Approximately a third of Gaza's malaria cases come from Chibuto and Limpopo districts. Despite the widespread distribution of LLINs every three years in these Gaza districts, malaria cases remain alarming despite the implementation of this initiative [10]. Several studies have been conducted to understand factors that influence LLIN usage to reduce malaria burden, but few have addressed the quality of key messages provided by net distributors in communities and health facilities (mass distributions of LLINs and antenatal consultations).

The aim of this study is to evaluate the factors associated with the use of LLINs (including key messages received) by pregnant women and women with children under 5 years old in Limpopo and Chibuto districts, Mozambique.

## Materials and methods

### Study design

A descriptive, qualitative cross-sectional study was conducted between 15th and 21st June 2022 (1-week period). An in-depth interview method was conducted with pregnant women and mothers with children under five years old, exploring their beliefs, experiences, and perception of messages conveyed by health professionals when LLINs were being supplied.

### Study site and population

The study was carried out in two rural districts of Gaza Province in Mozambique—Chibuto and Limpopo. With 154,121 residents, Limpopo district has seven primary health centres (PHCs, or health centres). Chibuto is home to 222,540 people and 19 PHCs. In both districts, agriculture, cattle raising, and fishing are the primary economic activities[11].

We randomly selected one high volume and one low volume PHC in each district. The number of monthly mother and child health (MCH) consultations was used to define volume, a high volume PHC would have $\geq$ 220 MCH consultations per month. Low volume PHC would have <150 MCH consultations per month. In Chibuto district the following centres were selected: Centro de Saúde de Chibuto and Centro de Saúde de Chaimite; and in Limpopo district: Centro de Saúde de Chicumbane and Centro de Saúde de Licilo, high and low volume respectively.

### Participant selection and sample size

Pregnant women were identified at the antenatal clinic (ANC). Mothers of children under 5 years were identified in the outpatient (OPD) consultations, in the post-natal consultations, the consultations of children at risk (CCR) and healthy children (CCS).

The study investigator explained the study and study procedures to all the women before their consultation. The clinician screened the women for eligibility to the study during the consultation (see **Table 1**), and only those who were eligible were recruited and sent to the investigator. Following that, each woman was interviewed individually in a private consultation room at the PHC by the study investigator.

In both groups, convenient non-probabilistic sampling was used based on the availability of potential study participants and their consent. An overall sample size of 48 women was determined by data saturation; 24 in Chibuto (14 in CS Chibuto, 10 in CS Chaimite) and 24 in Limpopo (14 in CS Chicumbane, 10 in CS Licilo) (see **Table 2**).

For a variety of reasons, we chose a convenient non-probabilistic sampling method: 1) cost concerns; the study did not have any funding, and random sampling would have been more expensive in such a large and dispersed population (time and personnel costs, rental of interview recording devices, transport costs), 2) The study was time sensitivity to provide quick answers to a programmatic issue on the ground; 3) this was an exploratory study aimed at generating hypotheses for future studies using random sampling to be more comprehensive.

**Table 1. Inclusion and exclusion criteria.**

| Group | Inclusion criteria | Exclusion criteria |
|---|---|---|
| Pregnant women | • Pregnant women $\geq$ 18 years in the ANC<br>• Given informed consent for the study | • Pregnant women with mental health disorder |
| Mothers of children under 5 years old | • Mothers $\geq$18 years old with children under 5 years old coming for consultation at the PHC<br>• Given informed consent for the study | • Women accompanying children under 5 years who are not their actual mothers |

**Table 2. Distribution of the sample size per PHC.**

| District | PHC | Sample | |
|---|---|---|---|
| **Chibuto** | **CS Chibuto** | Pregnant women | 7 |
| | | Mothers with children under 5 years | 7 |
| | **CS Chaimite** | Pregnant women | 5 |
| | | Mothers with children under 5 years | 5 |
| **Limpopo** | **CS Chicumbane** | Pregnant women | 7 |
| | | Mothers with children under 5 years | 7 |
| | **CS Licilo** | Pregnant women | 5 |
| | | Mothers with children under 5 years | 5 |
| **Total** | | | **48** |

During the study, we included women who presented at the selected health centres with characteristics of interest. To determine sample size, we used theoretical or data saturation, but with the predicted minimum of 14 women in high-volume health centres and 10 in low-volume health centres. In this study, we were able to saturate the information without exceeding the minimum amount predicted.

## Procedure

During morning health talks conducted at both health facilities, study investigators identified participants in both groups. Potential participants were informed of the study's procedures and importance. After their routine consultations, they were informed that if they wished to participate, they would be accompanied by a study team member to an in-depth interview room after giving their informed consent.

Three well-trained study surveyors conducted in-depth interviews in a separate consultation room so as to protect patient privacy. Patients were interviewed individually, without the presence of family members or friends. A 45-minute interview, including informed consent, was conducted for each person. An average of four interviews were conducted per day by each surveyor; the interviews were completed within 1-week period (4 days at high volume PHCs and 3 days at low volume PHCs). We conducted the interviews in Portuguese or Changana (local languages) and recorded them. Following transcription and translation (for interviews conducted in Changana), the interviews were analysed. Translations were carried out by one of the study surveyors and double checked by the principal investigator.

All participants were interviewed using an interview guide (S1 Text). It included closed questions to characterize them and open questions to determine their interests. The interview guide was study specific and has not been utilized in a previous study.

As women presented at the consultation and an investigator was available to interview them, they were recruited. If all investigators were conducting interviews, some women might have come for consultation and left without being interviewed. A daily average of 10–15 pregnant women and 18–25 mothers with children under 5 years attend the health centres for consultation.

## Data management and analysis

After recording the in-depth interviews on paper, the transcripts were transferred to Microsoft Word (2010) with questions and responses arranged by participant placement. There were two databases created, one for pregnant women and one for women with children under five. The instruments and informed consents used in the study were kept in a secure locker at the Gaza

Provincial Health Directorate, with only the principal investigator having access. The archive will be stored for 5 years before being destroyed.

The transcripts were read by three investigators (AVN, AM and IC). Based on the discussion of patterns of facts and events that regularly came up within and between the groups, the three investigators constructed emerging themes. A set of codes along with their respective guidelines for applying them were developed in response to these emerging themes. Each investigator coded the transcripts independently, and they held regular meetings to discuss and reconcile their respective codes to ensure agreement. Coding the transcripts and extracting coding reports were done using Maxqda software, version 12, which was then used to prepare the data reduction and summary tables. In the final phase of data analysis, all researchers discussed the results. Analysis, interpretation of the data and elaboration of the final report were conducted between July 2022 and February 2023.

### Ethics statement

This study was conducted according to the Declaration of Helsinki principles and obeyed Good Clinical Practice (GCP) guidelines. Ethical approval (for protocol and written informed consent documents) was granted by the Institutional Review Board of the Faculty of Health and the Central Hospital of Maputo, the Gaza Institutional Ethics Committee and the scientific council of the Faculty of Health of the University of Eduardo Mondlane. Considering that the study involved vulnerable populations, it was also submitted and approved by the National Health Bioethics Committee (CNBS) in Mozambique (ref # 34/CNBS2022).

## Results

58.3% of the women interviewed were between 18 and 27 years old. In both groups of women, majority had a primary school education (52.1%), were married or in a de facto union (79.2%) and were unemployed (89.6%). Sociodemographic characteristics are summarized in Table 3.

According to Table 4, nine themes emerged with their respective codes to standardize the data analysis. We identified the following themes: 1) perceptions of LLIN use, 2) awareness of vulnerable groups in malaria prevention by mothers, 3) misconceptions about LLIN use, 4) barriers to personal use of LLINs, 5) barriers to the use of LLINs within communities, 6) seasonal preferences for LLIN use, 7) health professionals' key messages during the distribution of LLINs in health centres, and 8) key messages from health professionals during community distribution of LLINs. In addition, factors that facilitate the use of LLINs were discussed with participants.

LLINs are perceived to prevent mosquito bites and malaria by a majority of pregnant women (PW; 83.3%) and mothers with children under 5 years (MCU5; 75.0%). LLINs are believed to protect against other diseases such as cholera by 16.7% of individuals in each of the two groups, and a minority of MCU5 believes LLINs are useful for fishing.

79.2% of MCU5 are aware that pregnant women and children are vulnerable groups in malaria prevention compared with 54.2% of PW. Both groups believe all groups should be prioritized, including pregnant and postpartum women under 25.

The majority of women in both groups (87.5% PW and 100% MCU5) reported no myths or taboos about LLINs. Three pregnant women report misperceptions about LLINs increasing mosquito populations and spreading diseases. MCU5 (54.2%) reported no side effects or problems associated with personal use of LLINs. In contrast, 83.3% of PW reported side effects from LLIN usage. Four women reported difficulty breathing while using LLINs in both groups. In five of the MCU5, alternative methods of prevention were used instead of LLINs (such as sprays or plants with mosquito repellent effects). In response to a question regarding barriers

**Table 3. Socio-demographic characteristics of participants.**

| Characteristics | Frequency (%) | |
|---|---|---|
| | Mothers with children under 5 years old | Pregnant women |
| *Age* | | |
| 18–22 | 12 (50.0) | 4 (16.7) |
| 23–27 | 5 (20.8) | 7 (29.2) |
| 28–32 | 2 (8.3) | 6 (25.0) |
| 33–37 | 2 (8.3) | 4 (16.7) |
| 38–42 | 2 (8.3) | 2 (8.3) |
| ≥ 43 | 1 (4.2) | 1 (4.2) |
| | | |
| *Education level* | | |
| No formal education | 3 (12.5) | 4 (16.7) |
| Primary | 14 (58.3) | 11 (45.8) |
| Secondary | 7 (29.2) | 9 (37.5) |
| | | |
| *Marital status* | | |
| Single | 3 (12.5) | 6 (25.0) |
| Married/Union | 20 (83.3) | 18 (75.0) |
| Widow | 1 (4.2) | 0 |
| Divorced/separated | 0 | 0 |
| | | |
| *Occupation* | | |
| Stay-at-home (no paid activity reported) | 22 (91.7) | 21 (87.5) |
| Employed (incl. civil servants) | 1 (4.2) | 1 (4.2) |
| Self-employed | 1 (4.2) | 2 (8.3) |

to LLIN usage in the community, 66.7% of the PW and 91.7% of the MCU5 reported having heard that community members had experienced side effects from LLINs.

A majority of MCU5 (70.8%) reported using LLINs year-round, with no seasonal preference. In contrast, 58.3% of PW stated they preferred to use LLINs in the summer because they believed winter was mosquito-free. Most MCU5 (70.8%) reported not receiving any information at the health centres when they received LLINs, compared to 45.8% of PW. In only 4 MCU5, the correct message about what to do before using LLINs. According to four PW, they were only told not to use LLINs for fishing or to cover flowerbeds. As part of the community mass distribution, 62.5% of MCU5 and 50% of PW stated they had not received information regarding LLIN care before usage. Similarly, only 4 MCU5 received the correct message on what to do before using LLINs.

During the survey, women in both groups reported that a health professional's explanation of how to use and care for LLINs at the health centres or during mass distribution was very important to them. Participant suggestions included monitoring LLIN usage and delivering health talks on LLINs and malaria prevention in schools and communities on top of the distribution of LLINs. According to some women, to be able to get the net for free from the government is a good thing, as they could not afford it.

"*even now they gave us the net, but they didn't explain to me what I should do before starting to use it, so the Ministry of Health should review this and tell the nurses as well as the distributors of mosquito nets in the community to give a good explanation of the use of mosquito nets. Others open it and use it right away and suffer from allergies, so they may decide not to use the mosquito net anymore after this*" (PW number 10, 37 years old)

**Table 4. Definitions, codes and main results per emerging themes.**

| Theme | Definition | Codes | Pregnant women* (N = 24) n/ (%) | Mothers of children under 5years* (N = 24) n/ (%) |
|---|---|---|---|---|
| Perceptions of LLIN use | Emerged when the interviews discussed:<br>• The importance of using LLINs to protect themselves from malaria<br>• The use of LLINs by interviewees, both those who mentioned they were using them appropriately and inappropriately<br>• The use of LLINs by neighbours and acquaintances, both appropriately and inappropriately | Prevent mosquito bites and malaria | 20 (83.3) | 18 (75.0) |
| | | Prevent other diseases such as cholera | 4 (16.7) | 4 (16.7) |
| | | Protect the foetus | 0 | |
| | | Use o LLINs for fishing by others | 0 | 2 (8.3) |
| Awareness of vulnerable groups in malaria prevention by mothers | Emerged when interviewees discussed:<br>• Groups traditionally considered vulnerable in malaria prevention<br>• Reasons why pregnant women and children under 5 years old cannot be the only group considered priority in the use of LLINs. | Pregnant women and children | 13 (54.2) | 19 (79.2) |
| | | No specific risk group | 7 (29.2) | 5 (20.8) |
| | | Pregnant women and postpartum women < 25years old | 2 (8.3) | 0 |
| Misconceptions about LLIN use | Emerged when interviewees discussed:<br>• Interviewees beliefs regarding the use of LLINs<br>• Interviewees and community leaders' opinions on the use of LLINs<br>• Rumours associated with the use of LLINs | Increase in number of mosquitoes | 3 (12.5) | 0 |
| | | Increase in other diseases | | 0 |
| | | No myths or taboos heard | 21 (87.5) | 24 (100.0) |
| Barriers to personal use of LLINs | Emerged when interviewees discussed the difficulties that they have in utilizing LLINs adequately | No problems experienced | 0 | 13 (54.2) |
| | | Experienced side effects (e.g. phobia, allergy and increase in temperature) | 20 (83.3) | 4 (16.7) |
| | | Difficulty breathing | 4 (16.7) | |
| | | Discomfort in getting in and out of the LLINs | 0 | 2 (8.3) |
| | | Use of alternative prevention methods | 0 | 5 (20.8) |
| Barriers to the use of LLINs within communities | This theme was created in response to the interviewees discomfort in speaking about their own difficulties with LLINs use but were reporting that other people had mentioned having problems such as allergies, increase temperature when using LLINs | Experienced side effects | 16 (66.7) | 22 (91.7) |
| | | Alcoholic habits and perception that LLINs are ineffective | 2 (8.3) | |
| | | Dislike LLINs | 6 (25.0) | 0 |
| | | Fell ill with the flu due to LLIN use | | 2 (8.3) |
| Seasonal preferences for LLIN use | Emerged when interviewees discussed:<br>• Preferential use of LLINs in the winter<br>• Preferential use of LLINs in the summer<br>• Use of LLINs in both winter and summer<br>• Reasons behind the preferential use of LLINs in one season vs another | Prefer to use LLIN in summer because there are no mosquitoes in winter | 14 (58.3) | 3 (12.5) |
| | | Prefer to use LLIN in summer and winter alike | 10 (41.7) | 17 (70.8) |
| • Health professionals' key messages during the distribution of LLINs in <u>health centres</u> | Emerged when interviewees discussed:<br>• Information received on the importance of LLINs use<br>• Precautions to be taken before LLINs use<br>• Not having received any information about LLINs use<br>• Having received correct information on the precautions to be taken before LLINs use<br>• Having received incorrect information on how to care for LLINs before use | No information given | 11 (45.8) | 17 (70.8) |
| | | Wash LLIN with detergent and/ or dry it under the sun | 5 (20.8) | 1 (4.2) |
| | | Wash LLIN with water and dry it in the shade | 5 (20.8) | 0 |
| | | Air LLIN in the shade, before use | 0 | 4 (16.7) |
| | | Told not to use for fishing or cover flowerbeds | 4 (16.7) | 0 |

(*Continued*)

**Table 4.** (Continued)

| Theme | Definition | Codes | Pregnant women* (N = 24) n/ (%) | Mothers of children under 5years* (N = 24) n/ (%) |
|---|---|---|---|---|
| Key messages from health professionals during community distribution of LLINs | Emerged when interviewees discussed:<br>• Information received on the importance of LLINs use<br>• Precautions to be taken before LLINs use<br>• Not having received any information about LLINs use<br>• Having received correct information on the precautions to be taken before LLINs use<br>• Having received incorrect information on how to care for LLINs before use | No information given | 12 (50.0) | 15 (62.5) |
| | | Wash LLIN with detergent and/or dry it under the sun | 5 (20.8) | 2 (8.3) |
| | | Wash LLIN with water and dry it in the shade | 0 | 0 |
| | | Air LLIN in the shade, before use | 0 | 4 (16.7) |
| | | Told not to use for fishing or cover flowerbeds | 4 (16.7) | 0 |

*In cases where responses exceeded the number of women in the group, some women provided responses that fit into more than one category. In cases where the total number of responses is less than the total number of women, some women declined to answer or did not know what the answer was.

"*Health authorities should also use schools, our local chiefs so that they can teach us how to use to mosquito nets correctly, and monitor the use in the community so that people are not using it in the mangroves and for fishing.*" (MCU5 number 20, 18 years old)

"*The government must always distribute mosquito nets because, for example, they are very expensive for the money we make and it can happen that others don't have the money to buy mosquito nets and die of malaria*" (PW number 10, 37 years old)

## Discussion

A majority of women in both study groups knew that LLNIs prevented mosquito bites, which in turn prevented malaria. The findings are consistent with those of other African countries (South Africa and Ethiopia), where pregnant women and mothers with young children have stated that LLINs provide a physical barrier between them and mosquitoes, decreasing mosquito bite risk. According to the same studies, LLINs also disorient or kill resting mosquitoes by their repellent effect [12,13].

According to a study in Malawi, respondents did not reveal the truth because they feared that the government would paralyze net distribution. Rather than focusing on what respondents knew about LLINs, they sought to learn what other community members knew. As a result of drought and food insecurity, people believed they could sell or use the nets for fishing in order to increase family income [7]. A number of other studies have also reported adverse environmental and economic effects of the widespread use of LLNIs for fishing in East Africa and northern Mozambique [8,14]. We did not find this to be a major finding in our study even though fishing is a primary economic activity in these communities. This is despite the fact that participants were questioned about their friends and community members' opinions when they showed reluctance to answer questions. Only a few (less than 10%) reported that community members used LLNIs for fishing or covering flowerbeds.

For the majority of women in both study groups, pregnant women and young children were identified as the most vulnerable populations in need of priority attention for malaria prevention. This is aligned with a study conducted in Ethiopia which reported that the population understood the importance of protecting children and pregnant women [15]. As in a similar study conducted in Madagascar, we found that 50% of the women in our study felt that other vulnerable groups living around them should be equally protected [16]. Researchers in

the Central region of Mozambique found that school-aged children are at greater risk of malaria infection and severe disease [17].

A small portion of pregnant women (12.5%) believed LLNIs would increase the mosquito population and spread diseases. This is in line with one study in Ethiopia, where respondents believed that LLINs breed bed bugs (small parasitic insects that feed exclusively on humans and domestic animals' blood) [12]. There were no misconceptions reported by respondents in the MCU5 group. Possibly, pregnant women may be more sensitive to anything that may harm their unborn children, making them vulnerable to misinformation.

While most MCU5 did not report any problems with LLINs, 100% reported experiencing side effects and difficulty breathing from using LLINs during pregnancy. A number of studies have identified side effects of LLINs, particularly if they are not used properly. Among other things, users may experience itchiness, heat, discomfort, and respiratory problems [18–21]. Pregnant women might experience side effects from LLINs due to increased sensitivity during pregnancy and possible reactions to the chemicals in the nets, which could account for the discrepancy between PW and MCU5. However, when asked about the community 91.7% of MCU5 reported having heard about others experiencing side effects with LLIN usage. A reluctance to answer the question honestly might also have played a part. According to a study in Ghana, community members used alternative methods of malaria prevention, such as herbs, grass, burning range peels, or sprays [19]. In our study, few MCU5 reported using alternative methods of mosquito prevention. These products are believed to contain effective insect repellents.

The study groups differed in their seasonal preferences. Majority of pregnant women preferred to use the LLIN in summer. According to studies conducted in Myanmar and Madagascar, women use LLIN more in summer because they believe there is no mosquito activity in winter [16,22]. A study in Nigeria found, however, that pregnant women did not use LLINs in summer due to the higher temperatures [23]. As mosquitoes are observed at all times of the year, 70.8% of the MCU5 reported using LLINs throughout the year, in line with findings from other studies [24].

Health professionals should inform the targeted population of what to do before using LLINs and how to care for them once they receive LLINs from health centres or in the community. Communication should focus on how LLINs should be used, installed, maintained, handled, how to deal with side effects, when to replace, and special advice for pregnant women [25]. We found that both groups of women received insufficient or no information in our study. Among those who received information about what to do before using the nets, only 16.7% received the correct information. The lack of knowledge about the proper use and care of LLINs may hinder their utilization and increase malaria transmission. As a result, it is not surprising that when women were asked how LLINs can be used more effectively, they suggested getting better information about how to use and care for them, monitoring the use and care of them in the community, and engaging the community by delivering health talks at schools and community gathering places.

## Limitations

The study had some notable limitations. Women were hesitant to participate in the study, because participation meant staying for a longer period than initially planned in the health centre. To minimize this limitation, during the health talks in the waiting rooms the women were informed beforehand about the study and its procedures. Observer bias was an additional limitation and to address this the investigators used follow up questions whenever necessary. Women were also hesitant about participating in the study in fear of suffering retaliation and

being excluded from future LLINs distribution campaigns. These fears were also addressed during the health talks and before the interviews, to ensure they understood that their responses would not influence current or future healthcare provision and that they were free to stop their participation anytime they felt uncomfortable.

Additionally, there are several limitations and potential sources of bias associated with convenient non-probabilistic sampling, which may affect the validity and generalizability of the findings. Hence, results should be interpreted with caution since it does not give a representative sample of the larger population.

## Conclusion

Providing clear, culturally sensitive, and practical information on the correct use of LLINs, as well as regular monitoring of their proper use, would be a great step forward for Mozambique's national malaria program. Communities will be empowered to minimize risks and maximize effectiveness in preventing malaria.

## Supporting information

**S1 Data. Demographic data.**
(XLSX)

**S1 Text. Interview guide.**
(DOCX)

**S2 Text. Transcriptions PW.**
(DOCX)

**S3 Text. Transcriptions WCU5.**
(DOCX)

## Acknowledgments

The authors would like to thank the women who participated in the study, the data collectors and the health professionals in the facilities selected for the study.

## Author Contributions

**Conceptualization:** Amancio Vicente Nhangave, Abuchahama Saifodine.

**Data curation:** Amancio Vicente Nhangave, Alfa Moiane, Ismail Chiposse.

**Formal analysis:** Amancio Vicente Nhangave, Alfa Moiane, Ismail Chiposse, Izaidino Muchanga, Abuchahama Saifodine.

**Investigation:** Isabelle Munyangaju, Dulce Osório, Edy Nacarapa, Sozinho Ndima, Alfa Moiane.

**Methodology:** Amancio Vicente Nhangave, Dulce Osório, Edy Nacarapa, Sozinho Ndima, Ismail Chiposse, Izaidino Muchanga, Abuchahama Saifodine.

**Project administration:** Isabelle Munyangaju, Ismail Chiposse, Abuchahama Saifodine.

**Supervision:** Alfa Moiane, Izaidino Muchanga, Abuchahama Saifodine.

**Validation:** Izaidino Muchanga, Abuchahama Saifodine.

**Writing – original draft:** Amancio Vicente Nhangave, Isabelle Munyangaju.

**Writing – review & editing:** Amancio Vicente Nhangave, Isabelle Munyangaju, Dulce Osório, Edy Nacarapa, Sozinho Ndima, Izaidino Muchanga, Abuchahama Saifodine.

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
