## [Decision Letter · Decision Letter 0]

4 Oct 2023

PGPH-D-23-01427

Factors associated with the use of mosquito nets in pregnant women and mothers with children under five years of age in Gaza province, Mozambique

Dear Dr. Munyangaju,

Thank you for submitting your manuscript to PLOS Global Public Health. After careful consideration, we feel that it has merit but does not fully meet PLOS Global Public Health’s publication criteria as it currently stands. Therefore, we invite you to submit a revised version of the manuscript that addresses the points raised during the review process.

We look forward to receiving your revised manuscript.

Kind regards,

Abhinav Sinha, M.D.

Academic Editor

Journal Requirements:

2. We have noticed that you have uploaded Supporting Information files, but you have not included a list of legends. Please add a full list of legends for your Supporting Information files after the references list.

3. In the online submission form, you indicated that "The datasets analysed during the current study are available from the corresponding author on reasonable request". All PLOS journals now require all data underlying the findings described in their manuscript to be freely available to other researchers, either 1. In a public repository, 2. Within the manuscript itself, or 3. Uploaded as supplementary information.

Additional Editor Comments (if provided):

Reviewers' comments:

Reviewer's Responses to Questions

**Comments to the Author**

1. Does this manuscript meet PLOS Global Public Health’s publication criteria? Is the manuscript technically sound, and do the data support the conclusions? The manuscript must describe methodologically and ethically rigorous research with conclusions that are appropriately drawn based on the data presented.

Reviewer #1: Partly

Reviewer #2: Yes

2. Has the statistical analysis been performed appropriately and rigorously?

Reviewer #1: No

Reviewer #2: N/A

3. Have the authors made all data underlying the findings in their manuscript fully available (please refer to the Data Availability Statement at the start of the manuscript PDF file)?

Reviewer #1: Yes

Reviewer #2: Yes

4. Is the manuscript presented in an intelligible fashion and written in standard English?

Reviewer #1: Yes

Reviewer #2: Yes

5. Review Comments to the Author

Reviewer #1: The MS has a good question to understand the risk of pregnant ladies however few observations are seen

In Methods Residents living in both province are quite good in number why the convenient non-probabilistic sampling was done and only 48 pregnant and child bearing mother were taken in study which gives a lower confidence in the study authors should justify the sample size

Results As we are interested in the perception and not in the who gave the answers therefore these answers may be point wise or percent wise can be put in the MS rather using their answers

Put data in a paragraph form the answers by the respondent can be used in passive form and in percent perception etc.

The result section should have such paragraphs or results and percentages as written in discussion

In discussion authors mau consider writing a contentious discussion on the awareness and why the results were observed so

Reviewer #2: I find the research work to be highly significant and a valuable contribution to the field. This manuscript has the potential to garner great interest among readers, as it addresses the critical gap in the successful implementation of the most crucial intervention for combating malaria on a global scale. The topic itself is highly relevant, and the findings are indeed intriguing, possessing the capacity to capture the attention of stakeholders who can leverage these results to reshape policies and strategies for the improved utilization of LLIN/ITN (Long-Lasting Insecticidal Nets/Insecticide-Treated Nets).

However, there are several important issues that require the authors' attention:

The manuscript follows the 'Plos Global Public Health' standard, but there is room for improvement in the clarity of the results and discussion sections.

The manuscript's title is somewhat unclear, as it mentions mosquito nets, while a portion of the manuscript specifically focuses on LLINs. Consider revising the title to reflect the broader scope of LLIN usage.

The title page of the manuscript lists authors in a different sequence than the submitted author list, and the affiliation of the corresponding author on the title page differs. These discrepancies need to be addressed.

The use of the terms "mosquito nets" and "LLIN" appears inconsistent throughout the manuscript. It is advisable to use a consistent term for all types of nets.

A thorough review of the English language and grammar is necessary, as some sentences are difficult to decipher, and punctuation issues arise, particularly in the results section.

The abstract format does not align with the 'Plos Global Public Health' guidelines, and it should not contain subheadings. Additionally, consider shortening the conclusion section by one or two lines.

In the introduction section:

Properly format the reference "WH, 2022" in a standard referencing style.

Clarify whether Cabo Delgado is a state or district in line 3 of paragraph 2.

Replace the phrase "massive amount" with a more scientifically appropriate term, and specify the relevant authority's name in paragraph 3.

Clarify whether Mozambique ranks 11th in Africa or globally in paragraph 3, line 4.

Add the prefix "national" to the malaria control strategic plan's name in paragraph 5, line 1.

Consider mentioning studies from other African countries working on the same factors to highlight the existing research and identify gaps.

Clearly articulate the gap in knowledge if studies on the mentioned factors are available.

The study's aim should include the country name, as mentioned in the abstract.

In the "Materials and Methods" section:

Revise "in depth interview process" to "in-depth interview method."

Specify whether the study was conducted within a one-month or one-week time period.

Clarify whether healthcare facilities and health units are considered the same class of health centers.

Explain how the sample size remained consistent across both districts, considering potential data variations. It appears that the sample size was pre-determined, with interviews conducted accordingly based on the different categories.

Clarify whether the interviews were conducted individually or in the presence of family members.

Provide information on who translated the past questions and responses when the native language was used.

Specify how many individuals were contacted and how many agreed to participate in the study.

Elaborate on the timeline and individuals responsible for data analysis.

If the interview module used was developed in 2021 and used in the 2022 study, please reference the source of the interview module if it was utilized in a previous study.

In the "Results" section:

Address the discrepancy between the occupation categories and the module questions, particularly regarding the "stay at home" and "white stick whites" categories.

Provide a brief overview of the data presentation before presenting the actual data.

Consider including a table of themes and subthemes to enhance reader comprehension.

In paragraph 2, line 3, discuss the relevance of "Nets to fish and protect flower beds due to smell and increased heat" in the results section, as it was not previously mentioned.

Discuss the study results in the context of previous research conducted on the same cohort in African and global populations.

6. PLOS authors have the option to publish the peer review history of their article (what does this mean?). If published, this will include your full peer review and any attached files.

**Do you want your identity to be public for this peer review?** For information about this choice, including consent withdrawal, please see our Privacy Policy.

Reviewer #1: No

Reviewer #2: No

---

## [Decision Letter · Decision Letter 1]

22 Nov 2023

PGPH-D-23-01427R1

Factors associated with the use of mosquito nets in pregnant women and mothers with children under five years of age in Gaza province, Mozambique

Dear Dr. Munyangaju,

Thank you for submitting your manuscript to PLOS Global Public Health. After careful consideration, we feel that it has merit but does not fully meet PLOS Global Public Health’s publication criteria as it currently stands. Therefore, we invite you to submit a revised version of the manuscript that addresses the points raised during the review process.

We look forward to receiving your revised manuscript.

Kind regards,

Abhinav Sinha, M.D.

Academic Editor

Journal Requirements:

Additional Editor Comments (if provided):

Thanks for addressing the comments. Would be happy to accept the paper after some minor corrections as suggested by the reviewer.

Reviewers' comments:

Reviewer's Responses to Questions

**Comments to the Author**

1. If the authors have adequately addressed your comments raised in a previous round of review and you feel that this manuscript is now acceptable for publication, you may indicate that here to bypass the “Comments to the Author” section, enter your conflict of interest statement in the “Confidential to Editor” section, and submit your "Accept" recommendation.

Reviewer #1: All comments have been addressed

2. Does this manuscript meet PLOS Global Public Health’s publication criteria? Is the manuscript technically sound, and do the data support the conclusions? The manuscript must describe methodologically and ethically rigorous research with conclusions that are appropriately drawn based on the data presented.

Reviewer #1: Yes

3. Has the statistical analysis been performed appropriately and rigorously?

Reviewer #1: No

4. Have the authors made all data underlying the findings in their manuscript fully available (please refer to the Data Availability Statement at the start of the manuscript PDF file)?

Reviewer #1: Yes

5. Is the manuscript presented in an intelligible fashion and written in standard English?

Reviewer #1: Yes

6. Review Comments to the Author

Reviewer #1: Most of the comments are addressed authors re check the answer that cost effectivity was issue in comment one by reviewer 1 and change it in MS as well funding limitations should not be depicted in interview based cross-sectional study

how does the it affects the cost if more samples were taken as it was only indepth interview only and was cross-sectional only

7. PLOS authors have the option to publish the peer review history of their article (what does this mean?). If published, this will include your full peer review and any attached files.

**Do you want your identity to be public for this peer review?** For information about this choice, including consent withdrawal, please see our Privacy Policy.

Reviewer #1: No

---

## [Editor Report · Decision Letter 2]

20 Dec 2023

Factors associated with the use of long-lasting insecticidal nets in pregnant women and mothers

with children under five years of age in Gaza province, Mozambique

PGPH-D-23-01427R2

Dear Dr Isabelle Munyangaju,

We are pleased to inform you that your manuscript 'Factors associated with the use of long-lasting insecticidal nets in pregnant women and mothers

with children under five years of age in Gaza province, Mozambique' has been provisionally accepted for publication in PLOS Global Public Health.

Best regards,

Abhinav Sinha, M.D.

Academic Editor